# Anthropometric Variables as Predictors of Semen Parameters and Fertility Outcomes after Varicocelectomy

**DOI:** 10.3390/jcm9041160

**Published:** 2020-04-18

**Authors:** Ramy Abou Ghayda, Robert Zakhia El-Doueihi, Jun Young Lee, Muhammad Bulbul, Nassib Abou Heidar, Jad Bulbul, Samer Asmar, Sung Hwi Hong, Jae Won Yang, Andreas Kronbichler, Jae Il Shin

**Affiliations:** 1Division of Urology, Brigham and Women’s Hospital and Harvard Medical School, Boston, MA 02115, USA; ramy.aboughayda@gmail.com; 2Department of Global Health and Population, Harvard T.H. Chan School of Public Health, Boston, MA 02115, USA; hong@hsph.harvard.edu; 3Division of Urology, American University of Beirut Medical Center, Beirut 1107 2020, Lebanon; robertdweihy@gmail.com (R.Z.E.-D.); mb30@aub.edu.lb (M.B.); na192@aub.edu.lb (N.A.H.); 4Department of Nephrology, Yonsei University, Wonju College of Medicine, Wonju Kangwon 26426, Korea; junyoung07@yonsei.ac.kr (J.Y.L.); kidney74@yonsei.ac.kr (J.W.Y.); 5Department of Radiology, Loyola University Medical Center, 2160 South 1st Avenue, Maywood, IL 60153, USA; jadbulbul@gmail.com; 6Surgery Department, Post-Doctoral Research Fellow at University of Arizona, Tucson, AZ 85721, USA; samerasmar94@surgery.arizona.edu; 7Department of Internal Medicine IV (Nephrology and Hypertension), Medical University Innsbruck, 6020 Innsbruck, Austria; andreas.kronbichler@i-med.ac.at; 8Department of Pediatrics, Yonsei University College of Medicine, Yonsei-ro 50, Seodaemun-gu, C.P.O. Box 8044, Seoul 03722, Korea

**Keywords:** body mass index, varicocele, varicocelectomy

## Abstract

Varicocele is the most common correctable male infertility factor and varicocelectomy has been a mainstay in the management of infertility. However, the role of varicocelectomy as a treatment option has been controversial, and the scientific debate around it is still ongoing. Our study aimed to explore the role of anthropometric variables of infertile patients and their relation to sperm parameters following varicocelectomy. The outcome of 124 infertile patients who underwent open sub-inguinal varicocelectomy by a single surgeon over the last ten years was studied. Post varicocelectomy, four semen parameters (volume, total count, motility, and morphology) were analyzed and adjusted according to anthropometric variables including age, varicocele grade, and body mass index (BMI) of patients. Total count and motility were significantly improved after surgery. Varicocelectomy improved semen parameters, notably the count and the motility, especially in younger patients, lower grades of varicocele patients, and low BMI patients. In addition, BMI was positively correlated with volume in pre-varicocelectomy and post-varicocelectomy.

## 1. Introduction

Infertility affects 15% of all couples who are intending to conceive [1]. Male factors alone contribute to 20–30% of all infertility cases, and when combined with the female element, it is responsible for 50% of overall cases [2]. Varicocele is known to be the most common correctable male infertility etiology [3]. Several studies have reported the prevalence of varicocele as 15–20% in healthy subjects, and 40–70% in men with infertility [4,5]. The exact pathophysiological mechanism explaining how varicocele affects testicular function is still uncertain. However, several studies have reported that a reduced testicular volume, weak sperm parameters, and changes in testosterone levels in infertile men have been observed in patients with varicocele [6].

Therefore, varicocele correction has been a mainstay in the treatment of male infertility. The current guidelines recommend treatment of varicocele should be performed in cases of documented infertility, in the presence of one or more abnormal semen parameters of sperm function tests, in palpable varicocele, and in normal female fertility or potentially correctable female fertility [7]. However, the efficacy of varicocele repair in infertile males has been controversial. Improvement in sperm parameters and testicular volume after varicocele repair has been reported in the literature, yet its correlation with spontaneous pregnancy has been unclear [8,9]. 

The prevalence of varicocele increases with decreasing BMI (body mass index)and is associated with adipose tissue compression of the left renal vein [10]. Although there are many studies on the correlation between BMI and the prevalence of varicocele [11], there are no studies that have compared semen parameters according to BMI after varicocelectomy [4,6,12]. Our study aimed to compare pre- and post-operative semen parameters and testicular volume according to anthropometric parameters following sub-inguinal varicocelectomy performed consistently by the same surgeon over ten years, and to assess post-operative fertility concerning changes in the sperm parameters and testicular volume. 

## 2. Methods

### 2.1. Study Populations

Data were collected from clinical charts of a single surgeon, and a retrospective data analysis was done in a single center in Lebanon [13]. The person responsible for the data collection included all the charts of infertile males with varicocele (clinically proven unilateral or bilateral varicocele with abnormal semen parameters), who had undergone varicocelectomy between 2007 and 2016, incorporating a total of 124 patients in the study. Demographic and anthropometric data regarding the patients, including, but limited to, age, hormonal profile, testicular size, physical exam, ultrasound results, co-morbidities, and sperm analysis pre- and post-varicocelectomy were collected and tabulated in an Excel sheet. Post-varicocelectomy semen analysis was taken systematically 2 months postoperatively. However, we did not follow up with patients who had multiple semen analyses on follow up. Patients were then grouped according to age, grade of the varicocele, and BMI. Patients with previous pelvic or urologic surgeries, who might have been at risk of developing obstructive azoospermia, were excluded. This study was approved by the American University of Beirut Medical Center’s Institutional Review Board (ID: SUR.MB.05) on 21 June 2017. Our study was conducted according to the ethical standards laid down in the 1964 Declarations of Helsinki and its later amendments.

### 2.2. Statistical Analysis

Exploratory data analysis (EDA) was carried out describing the central tendency (e.g., mean, median) and distribution (e.g., standard deviation, 95% confidence interval) of the various variables. A paired t-test was used to compare the variables before and after surgery. One-way ANOVA was also used to describe the distribution of a continuous measure according to different categories (age groups or varicocele grade). Scheffè analysis was done as a post-hoc analysis if indicated. If the number of patients in each group did not exceed twenty subjects, analysis was performed by a nonparametric test (Wilcoxon signed-rank test). We categorized the patients into two groups (high BMI and low BMI) based on the median value of the BMI level (28.6) because of the limited number of patients. Spearman correlation analysis was performed to analyze the correlation between BMI and other continuous parameters. All statistical analyses were performed with the IBM Statistical Package for the Social Sciences (SPSS), version 23.0 (IBM Corporation, Armonk, NY, USA). All differences were considered statistically significant at a *p*-value < 0.05.

## 3. Results

### 3.1. Characteristics of the Patients

A total of 124 men were evaluated for pre- and post-operative parameters following varicocelectomy. The demographics and features of the patients are presented in Table 1, which show that 83.8% of patients were under 40 years, and 76% and 70.5% of patients had grade I or II varicocele, respectively, on physical exam and ultrasound. 

### 3.2. Pre- and Post-Varicocelectomy Semen Parameters

The total count and the motility showed a statistically significant improvement after operation (Table 2). This improvement was observed in both groups, the high BMI group (total count: 16.26 ± 19.25 million/mL vs. 22.17 ± 22.99 million/mL, *p* = 0.018; and motility: 38.44% ± 19.67% vs. 43.85% ± 19.07%, *p* = 0.026), and the low BMI group (total count: 17.07 ± 17.81 million/mL vs. 29.50 ± 43.70 million/mL, *p* = 0.010; and motility: 36.77% ± 18.47% vs. 42.95% ± 18.27%, *p* = 0.002). In subgroup analysis, both groups showed no significant improvement in volume and morphology. 

### 3.3. Subgroup Analysis of Pre- and Post-Varicocelectomy Semen Parameters

When dividing the population according to age groups, the improvement in the total count was most significant in the group of patients <30 years, increasing from 19.28 ± 19.08 million/mL to 31.81 ± 43.87 million/mL (*p* = 0.018), and a significant change was also evident among patients aged between 30 and 40 years, with an increase from 13.92 ± 15.89 million/mL to 21.30 ± 25.24 million/mL (*p* = 0.016). As for motility, there was a statistically significant increase across all age groups; however, the highest age group (40–49 years) showed the greatest increase from 34.61% ± 19.91% to 43.56% ± 19.03% with a *p*-value of 0.048 (Table 3). 

The total count after varicocelectomy improved in grade I varicocele patients (on ultrasound), from 15.70 ± 14.67 million/mL to 23.80 ± 27.13 million/mL (*p* = 0.05) and in grade II patients from 18.04 ± 18.52 million/mL to 28.02 ± 35.51 million/mL (*p* = 0.004). Motility also improved in grade I patients from 32.38% ± 18.28% to 41.86% ± 16.72% (*p* = 0.004), and in grade II patients from 39.20% ± 16.80% to 45.93% ± 18.37% (*p* = 0.004). In contrast, morphology only improved in grade I varicocele patients post-operatively, from 43.30% ± 18.77% to 50.71% ± 18.15% (*p* = 0.01) (Table 4). 

Spearman correlation analysis highlighted that pre-operative (rho = 0.233, *p* = 0.012) and post-operative (rho = 0.189, *p* = 0.043) volume was significantly correlated with BMI. However, other pre-operative and post-operative parameters (total count, motility, and morphology) did not correlate with BMI. 

In patients with an age below 30 years and a lower BMI, the total count was increased from 20.21 ± 17.28 million/mL to 38.69 ± 56.55 million/mL (*p* = 0.041), and this trend was also observed in the subgroup of cases with age under 30 years, low BMI, and grade II varicocele (*n* = eleven patients), with an increase from 8.36 ± 7.32 million/mL to 12.27 ± 8.16 million/mL (*p* = 0.046). In patients with an age between 30 and 40 years and a higher BMI, the total count increased from 14.47 ± 9.90 million/mL to 24.73 ± 28.94 million/mL (*p* = 0.045). Subgroup analysis revealed that motility increased in the age group of 30–40 years with a higher BMI, and grade II varicocele (*n* = six patients), from 29.17% ± 9.17% to 50.08% ± 20.10% (*p* = 0.044). In subjects with a lower BMI, motility increased after surgery from 37.15% ± 19.86% to 42.11% ± 19.85% (*p* = 0.049) (Table 5). Three men aged <30, 4 men aged between 30 and 40, and 3 patients aged between 40 and 49 had missing data from their charts, and subsequently were excluded from the analysis of semen parameters pre- and post-varicocelectomy according to the grade of the varicocele. 

Out of 102 patients who were actively trying to conceive, the post-operative pregnancy rate after undergoing a sub-inguinal varicocelectomy was 52/102 (50.98%); further semen analysis, however, was not included in the study design. The remaining patients were either lost to follow-up or not married. There was no significant correlation between fertility and any of the improvements in sperm parameters mentioned above post-operatively. Moreover, there was no significant difference in fertility across the different age groups or varicocele grades.

## 4. Discussion

Our study found a significant improvement in the total sperm count and the motility of the sperm post-varicocelectomy across the entire cohort. This result is in concordance with previous studies (including meta-analysis) that have shown similar improvement [4,6,14,15,16]. In our study, subgroup analysis, younger patients, and lower grade of varicocele were associated with the post-operative increase of semen parameters. In addition, BMI was positively correlated with semen volume regardless of operation. 

It is interesting to note that the statistically significant increase in total sperm count was only seen in the group of patients below 30 years of age. Other studies reported that total sperm count was decreased in varicocele patients, and that varicocelectomy improves total sperm counts [4,5]. Although the cut-off age was different, this result was in agreement with the conclusions of a study conducted in Austria, whereby younger patients benefit most from the procedure [6]. Unlike our results, this study revealed that patients showed increased total sperm count regardless of grades of varicocele. Our results also showed a significant increase in post-operative total sperm count in the grade I and II groups, although grade III patients showed the highest increased total sperm count with a borderline significance (*p* = 0.054). In our study, among patients in the group below 30 years, the total sperm count was increased after varicocelectomy only in the lower BMI group.

As for sperm motility, there was a significant increase in all three age groups. This result is in agreement with the results of meta-analyses [4,16]. However, those meta-analyses did not perform subgroup analysis. Meanwhile, in our research, the most significant increase of semen motility was shown in the third age group (40–49 years), with an 8.95% (from 34.61% to 43.56%) increase. This also agrees with another study, whereby the older aged population can still benefit from the procedure regarding sperm motility. However, that study grouped together patients with an age between 25 and 63 years [6]. Like total sperm count, grade III and IV patients did not significantly increase post-operative sperm motility.

In our study, the lower grades exhibited more improvement in semen motility and total counts after varicocelectomy. These results are in concordance with those described by Polito et al.’s study, where grade I varicoceles showed the most significant improvement in sperm density and motility [17]. However, their results were contradicted by those of Jungwirth et al., and Onozawa et al., which showed that the higher the grade of varicocele, the more significant the improvement in total sperm count and sperm density post-varicocelectomy [6,18]. Including our study, all studies had a small number of patients and did not adjust for other potential variables that improve semen parameters. Considering the relatively equal number of patients in each group [17], semen motility and total counts may improve more in lower grade varicocele, although more research is clearly needed to draw definite inferences.

There are several theories about the development of varicoceles, but the exact mechanisms remain to be elucidated. It is hypothesized that a “nutcracker effect”, or compression of the left renal vein, is pivotal in the pathogenesis of varicocele [19]. Low BMI patients, and decreased adipose tissue, especially between the superior mesenteric artery and aorta, prone affected patients to compression of the renal vein, which thus provokes development and recurrence of varicocele [11,20,21,22]. 

Cho et al. accumulated the data from 33 randomized control trials regarding the effect of varicocelectomy on pregnancy rate and the improvement of semen parameters as a potential first-line therapeutic option [8]. To recommend varicocelectomy as a standard of care, more data are needed to perform this treatment option for infertility with a high grade of evidence [23]. In line with this conservative statement, review articles show higher impotence rates in the treatment arm post-varicocelectomy, as opposed to the controls who did not undergo surgery [24,25,26].

It has been well established in the literature that increasing BMI decreases semen parameters, mainly sperm count, sperm concentration, and semen volume [27]. Our study showed that semen volume mainly decreases with increased BMI, and this is independent of varicocelectomy. An extensive review published by Guo et al. showed that semen volume is especially susceptible to BMI [27]; however, the reason is currently unknown and could have hormonal factors or biochemical causes that are yet to be revealed.

## 5. Conclusions

In conclusion, undergoing a varicocelectomy seems to improve the patient’s semen parameters, especially the count and the motility. Moreover, undergoing a varicocelectomy at a younger age might have the most significant impact on these parameters. The grade of the varicocele is also essential when it comes to treatment, as grades I and II show higher rates of improvement in semen parameters post varicocelectomy. However, our study showed that no correlation was observed between fertility and the increase in these parameters, requiring further studies with a larger sample size in the future, in order to obtain substantial and significant results.

Some challenges/limitations to our study include (1) incomplete data in the clinic charts, (2) no long-term follow up post-varicocelectomy in some cases, (3) small sample size, and (4) the absence of periodic spermograms postoperatively that would help in trending the changes of the parameters studied. 

## Figures and Tables

**Table 1 jcm-09-01160-t001:** Characteristics of patients receiving varicocelectomy.

Characteristics		Number of Patients (%)
Age (years)	<30	52 (41.9%)
	30–40	52 (41.9%)
	40–49	18 (14.4%)
	>50	2 (1.6%)
Varicocele grade on PEx	I	22/100 (22%)
	II	54/100 (54%)
	III–IV	24/100 (24%)
Varicocele grade on US	I	9/61 (14.8%)
	II	34/61 (55.7%)
	III–IV	18/61 (29.5%)
Side of varicocele	Left	44/90 (48.9%)
	Right	0/90 (0%)
	Bilateral	46/90 (51.1%)
Bodyweight	BMI < 28.6	60/121 (49.6%)
	BMI ≥ 28.6	61/121 (50.4%)

BMI: Body mass index; PEx: physical examination; US: Ultrasound.

**Table 2 jcm-09-01160-t002:** Pre- and post-varicocelectomy semen parameters (volume, total count, motility, and morphology).

Parameters	Pre-Operative	Post-Operative	*p*-Value
Volume (mL)	3.46 ± 1.63	3.51 ± 1.46	0.703
Total count (million/mL)	16.89 ± 18.20	26.18 ± 34.29	<0.001
Motility (%)	37.23 ± 19.13	43.23 ± 18.48	<0.001
Morphology (%)	47.22 ± 20.25	47.67 ± 22.04	0.864

**Table 3 jcm-09-01160-t003:** Pre- and post- varicocelectomy semen parameters by age groups.

Age	Parameters	Pre-Operative	Post-Operative	*p*-Value
<30	Volume (mL)	3.51 ± 1.55	3.70 ± 1.60	0.385
	Total count (million/mL)	19.28 ± 19.08	31.81 ± 43.87	0.018
	Motility (%)	38.04 ± 18.59	43.65 ± 16.89	0.016
	Morphology (%)	50.30 ± 18.98	53.85 ± 22.01	0.573
30–40	Volume (mL)	3.38 ± 1.72	3.29 ± 1.29	0.669
	Total count (million/mL)	13.92 ± 15.89	21.30 ± 25.24	0.016
	Motility (%)	38.17 ± 19.31	43.40 ± 19.49	0.028
	Morphology (%)	45.90 ± 22.53	41.86 ± 21.44	0.130
40–49	Volume (mL)	3.63 ± 1.65	3.72 ± 1.46	0.850
	Total count (million/mL)	20.19 ± 21.39	26.35 ± 23.62	0.216
	Motility (%)	34.61 ± 19.91	43.56 ± 19.03	0.048
	Morphology (%)	48.33 ± 14.32	51.58 ± 17.61	0.481

**Table 4 jcm-09-01160-t004:** Semen parameters pre- and post-varicocelectomy according to the grade of the varicocele.

Grade	Parameters	Pre-Operative	Post-Operative	*p*-Value
**I**	Volume (mL)	3.58 ± 1.66	3.70 ± 1.60	0.719
	Total count (million/mL)	15.70 ± 14.67	23.80 ± 27.13	0.050
	Motility (%)	32.38 ± 18.28	41.86 ± 16.72	0.004
	Morphology (%)	43.30 ± 18.77	50.71 ± 18.15	0.010
**II**	Volume (mL)	3.40 ± 1.93	3.19 ± 1.37	0.370
	Total count (million/mL)	18.04 ± 18.52	28.02 ± 35.51	0.046
	Motility (%)	39.20 ± 16.80	45.93 ± 18.37	0.004
	Morphology (%)	49.10 ± 16.50	47.23 ± 17.94	0.339
**III–IV**	Volume (mL)	3.33 ± 1.63	3.41 ± 1.20	0.809
	Total count (million/mL)	15.04 ± 13.73	33.27 ± 51.88	0.054
	Motility (%)	39.59 ± 18.51	41.82 ± 16.80	0.479
	Morphology (%)	54.41 ± 18.49	53.14 ± 15.58	0.731

**Table 5 jcm-09-01160-t005:** Semen parameters pre- and post-varicocelectomy according to age and BMI.

Group	Parameters	Pre-Operative	Post-Operative	*p*-Value
Age < 30 (number of patients)				
BMI ≥ 28.6 (20)	Volume (mL)	4.33 ± 1.70	4.17 ± 1.40	0.002
	Total count (million/mL)	16.42 ± 21.81	21.10 ± 15.52	0.258
	Motility (%)	39.0 ± 21.44	43.50 ± 16.47	0.243
	Morphology (%)	48.33 ± 23.38	65.83 ± 19.85	0.311
BMI < 28.6 (29)				
	Volume (mL)	2.86 ± 1.20	3.35 ± 1.72	0.112
	Total count (million/mL)	20.21 ± 17.28	38.69 ± 56.55	0.041
	Grade II	8.36 ± 7.32	12.27 ± 8.16	0.046
	Motility (%)	37.17 ± 17.72	43.79 ± 18.06	0.045
	Morphology (%)	52.00 ± 18.13	50.54 ± 21.20	0.810
Age: 30–40				
BMI ≥ 28.6 (21)	Volume (mL)	3.95 ± 1.93	3.61 ± 1.01	0.424
	Total count (million/mL)	14.47 ± 9.90	24.73 ± 28.94	0.045
	Motility (%)	41.76 ± 18.06	46.90 ± 18.34	0.305
	Grade II	29.17 ± 9.17	50.08 ± 20.10	0.044
	Morphology (%)	52.44 ± 15.39	45.22 ± 15.39	0.142
BMI < 28.6 (27)				
	Volume (mL)	2.91 ± 1.46	2.98 ± 1.29	0.770
	Total count (million/mL)	11.85 ± 17.05	17.53 ± 22.51	0.179
	Motility (%)	37.15 ± 19.86	42.11 ± 19.85	0.049
	Morphology (%)	39.67 ± 23.38	39.56 ± 25.15	0.977
Age: 40–49				
BMI ≥ 28.6 (12)	Volume (mL)	3.44 ± 0.78	3.74 ± 1.40	0.538
	Total count (million/mL)	15.87 ± 23.23	17.86 ± 19.18	0.814
	Motility (%)	39.83 ± 19.50	46.25 ± 22.07	0.211
	Morphology (%)	53.50 ± 10.49	54.90 ± 14.54	0.513
BMI < 28.6 (4)				
	Volume (mL)	4.62 ± 3.35	2.75 ± 1.19	0.180
	Total count (million/mL)	29.5 ± 19.62	43.75 ± 30.70	0.357
	Motility (%)	37.25 ± 17.97	42.50 ± 9.57	0.109
	Morphology (%)	39.25 ± 14.36	41.67 ± 23.63	0.999

High BMI: BMI ≥ 28.6, Low BMI: BMI < 28.6.

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
