# Peer review of "Anthropometric Variables as Predictors of Semen Parameters and Fertility Outcomes after Varicocelectomy"

_jcm, 2020, doi:10.3390/jcm9041160_

Round 1

Reviewer 1 Report

Varicocele is the most common correctable male infertility factor. The efficacy of varicocelectomy is still controversial. In this manuscript, the aims of the authors were to compare pre- and post-operative semen parameters and testicular volume according to the anthropometric variables of infertile patients and to assess post-operative fertility. The outcome of 124 infertile patients who underwent open sub-inguinal varicocelectomy by the same surgeon over the last ten years (2007-2016) was studied.

Four parameters (volume, total count, motility and morphology) were analyzed and adjusted according to anthropometric variables including age, varicocele grade and body mass index (BMI). The authors found that varicocelectomy improved semen parameters, notably sperm count and motility, especially in younger patients, patients with lower grades of varicocele and low BMI patients. No correlation was observed between fertility and the increase in these parameters. The authors also found that BMI was positively correlated with volume before and after varicocelectomy.

The authors acknowledge that the limitations of their study include (1) incomplete data in the clinic charts (2) no long-term follow up post-varicocelectomy in some cases (3) small sample size (4) absence of periodic spermograms postoperatively.

The manuscript is clear, well written and the data presented are interesting. Modifications are required to improve the manuscript:

- In this study, 52 patients aged <30 years, 52 patients aged 30-40 years and 18 patients aged 40-49 years were included. Please explain why the data of 49 patients aged <30 years, 48 patients aged 30-40 years and 16 patients aged 40-49 years are shown in Table 5.

- Please indicate how long after the surgery the sperm parameters were analyzed. Is there an improvement of sperm parameters shortly after the surgery? For the patients who benefited from a long-term follow-up, is there an improvement of semen parameters over time?

- The authors found that BMI was positively correlated with semen volume regardless of the operation. Please propose in the Discussion section a hypothesis to explain this positive correlation.

- Line 137: Please replace 50.8 by 50.08

Author Response

Reviewer:
- In this study, 52 patients aged <30 years, 52 patients aged 30-40 years and 18 patients aged 40-49 years were included. Please explain why the data of 49 patients aged <30 years, 48 patients aged 30-40 years and 16 patients aged 40-49 years are shown in Table 5.

REPLY: Thank you for your kind remarks. The missing patients in each age categories were excluded because of their incomplete data in their respective charts.  We added a sentence to reflect this in the results section line 139-142.

- Please indicate how long after the surgery the sperm parameters were analyzed. Is there an improvement of sperm parameters shortly after the surgery? For the patients who benefited from a long-term follow-up, is there an improvement of semen parameters over time?

REPLY: Thank you for this valuable comment. The above comments were taken into consideration in the revision and pertinent statements were added in the methods section in lines 73 and 74.

- The authors found that BMI was positively correlated with semen volume regardless of the operation. Please propose in the Discussion section a hypothesis to explain this positive correlation.

REPLY: Thank you for your valuable suggestions. A small dissertation about our findings was added at the end of the discussion section. Lines 195-200

- Line 137: Please replace 50.8 by 50.08

REPLY: Thank you, the amendments were made accordingly in the results section

Thank you for your esteemed reviews.

Best regards,

Ramy Abou Ghayda, MD

Reviewer 2 Report

The text is well written and the results are clear.

The subject is not very original since there are several papers regarding varicocelectomy and semen improvement, but the analyzed parameters are relevant.

Author Response

Manuscript ID: jcm-786500

Manuscript Title Anthropometric variables as predictors of semen parameters and fertility outcomes after varicocelectomy

Date: April 2020

Reviewer: The text is well written and the results are clear.

The subject is not very original since there are several papers regarding varicocelectomy and semen improvement, but the analyzed parameters are relevant.

REPLY

Thank you for your kind reviews.

Through our results, we learned that undergoing varicocelectomy at a younger age might have the most significant impact on sperm count, and motility. 

Best regards,

Jun Young Lee, MD

Reviewer 3 Report

Overall nice simple study with tangible action items.  The novelty is marginal in that we already know the poor effect of BMI on semen quality but the additional variable of varicocele is novel.

Author Response

Manuscript ID: jcm-786500

Manuscript Title Anthropometric variables as predictors of semen parameters and fertility outcomes after varicocelectomy

Date: April 2020

Reviewer: 

Overall nice simple study with tangible action items.  The novelty is marginal in that we already know the poor effect of BMI on semen quality but the additional variable of varicocele is novel.

REPLY

Thank you for your kind reviews.

Best regards,

Jun Young Lee, MD